# CEEMDAN-IPSO-LSTM: A Novel Model for Short-Term Passenger Flow Prediction in Urban Rail Transit Systems

**DOI:** 10.3390/ijerph192416433

**Published:** 2022-12-07

**Authors:** Lu Zeng, Zinuo Li, Jie Yang, Xinyue Xu

**Affiliations:** 1School of Electrical Engineering, Jiangxi University of Science and Technology, Ganzhou 341000, China; 2State Key Lab of Rail Traffic Control and Safety, Beijing Jiaotong University, Beijing 100044, China; 3Ganjiang Innovation Academy, Chinese Academy of Sciences, Ganzhou 341000, China

**Keywords:** urban rail transit, short-term passenger flow prediction, complete ensemble empirical mode decomposition with adaptive noise, long-short term memory neural network, improved particle swarm optimization, combination model, CEEMDAN-IPSO-LSTM

## Abstract

Urban rail transit (URT) is a key mode of public transport, which serves for greatest user demand. Short-term passenger flow prediction aims to improve management validity and avoid extravagance of public transport resources. In order to anticipate passenger flow for URT, managing nonlinearity, correlation, and periodicity of data series in a single model is difficult. This paper offers a short-term passenger flow prediction combination model based on complete ensemble empirical mode decomposition with adaptive noise (CEEMDAN) and long-short term memory neural network (LSTM) in order to more accurately anticipate the short-period passenger flow of URT. In the meantime, the hyperparameters of LSTM were calculated using the improved particle swarm optimization (IPSO). First, CEEMDAN-IPSO-LSTM model performed the CEEMDAN decomposition of passenger flow data and obtained uncoupled intrinsic mode functions and a residual sequence after removing noisy data. Second, we built a CEEMDAN-IPSO-LSTM passenger flow prediction model for each decomposed component and extracted prediction values. Third, the experimental results showed that compared with the single LSTM model, CEEMDAN-IPSO-LSTM model reduced by 40 persons/35 persons, 44 persons/35 persons, 37 persons/31 persons, and 46.89%/35.1% in SD, RMSE, MAE, and MAPE, and increase by 2.32%/3.63% and 2.19%/1.67% in R and R^2^, respectively. This model can reduce the risks of public health security due to excessive crowding of passengers (especially in the period of COVID-19), as well as reduce the negative impact on the environment through the optimization of traffic flows, and develop low-carbon transportation.

## 1. Introduction

The influence of human activities on the global climate, characterized by global warming, has had serious negative impacts on public health. Energy conservation and carbon reduction have become serious environmental development issues to address. At the 75th United Nations General Assembly on 22 September 2020, China announced it would reach a peak in CO_2_ emissions by 2030 and achieve carbon neutrality before 2060 (hereinafter referred to as double carbon goals) [1].

With the continuous improvement of China’s urbanization level and the diversification of urban transport logistics and travel demand, the transport sector has become the main body of China’s energy consumption and carbon emissions growth [2]. A key strategy for lowering urban carbon emissions is the expansion of public transportation [3,4]. Urban rail transit (hereinafter referred to as URT) is a large-capacity public transport infrastructure and the backbone of low-carbon transportation in cities. The URT in China has been rapidly increasing, and its energy consumption and carbon emission reduction pressure remains high. As of 30 September 2022, 52 mainland Chinese cities have put into operation 9788.64 km of URT lines, including 7655.32 km of subway, accounting for 78.21% [5]. Passenger flow volume is rapidly growing along with URT’s quick expansion, which is producing severe congestion in URT systems. Accurately predicting the short-term flow volume and subsequently carrying out the necessary management procedures are two ways by which to relieve traffic congestion [6,7]. Travelers can effectively change their preferred method of transportation, route, or travel dates in advance by properly forecasting the influx and outflow of each station in a URT, which reduces travel time and costs [8,9]. Utilizing the prediction data, operators can identify crowded stations. The relevant passenger control measures can be put in place at stations that are severely congested to prevent congestion. Moreover, the timetable can be timely optimized so as to transport more passengers during peak hours according to predictions results. 

At present, the research on short-time passenger flow prediction of URT at home and abroad is mainly conducted through three categories: statistical methods, traditional machine learning methods, and deep learning methods. Statistical methods are more sensitive to the linear relationship between variables, but they cannot capture the nonlinear relationship in the data. Such methods mainly include Kalman Filter model [10,11], ARMA model [12], and ARIMA model [13,14,15]. Traditional machine learning methods can better capture the nonlinear features in time series, and the accuracy for rail transit passenger flow prediction is higher. Such methods mainly include Support Vector Machine [16,17] and neural network [18,19,20]. However, the prediction model using traditional machine learning methods is prone to over-learning or under-learning problems when dealing with massive passenger flow data, which affects the accuracy of prediction models [21]. With the advancement of related theories and technologies, researchers have begun to use deep learning models to predict URT passenger flow [22]. Due to the strong applicability of the LSTM model in processing time series data, it has been widely used in passenger flow forecasting research [23,24,25]. 

The achievement of a single model’s good prediction performance in real-world case studies is undoubtedly difficult. As a result, more academics have increasingly concentrated on combination forecasting models. Gong et al. [26] set up a passenger flow forecasting framework combining the seasonal ARIMA-based method and Kalman filter-based method. The framework was applied to the real bus line for passenger flow prediction. Qin et al. [27] coupled a seasonal-trend decomposition approach with an adaptive boosting framework to anticipate the monthly passenger flow on China Railway. A prediction model for irregular passenger flow based on the combination of support vector regression and LSTM was presented by Guo et al. [28]. A three-stage passenger flow forecasting model was developed by Liu and Chen [29] using a deep neural network and stacked automated encoder. The performance of the prediction was shown to be significantly impacted by the choice and combination of important features.

Although the accuracy of the aforementioned prediction methods has somewhat increased, neither the interference of passenger flow data noise nor the manual trial-and-error method of determining the hyperparameters of the neural network based solely on empirical values has been considered. In order to address these issues, this paper combines the CEEMDAN algorithm for reducing data noise interference with the IPSO algorithm for hyperparameters optimization of LSTM neural networks to create a new short-term passenger flow prediction model of URT based on CEEMDAN-IPSO-LSTM. The model’s predictive performance is then thoroughly assessed using the benchmark function, prediction error, and Taylor diagram. In a word, short-term passenger flow accurate prediction of URT can improve the efficiency of transport infrastructure and means of transport. At the same time, it can further put forward optimization suggestions for URT operation management during the post-epidemic period, and provide a reference for the early realization of the dual carbon goals.

## 2. Methods

### 2.1. CEEMDAN Algorithm

The complete ensemble empirical mode decomposition with adaptive noise (CEEMDAN) algorithm is a time-frequency domain analysis method that excels at nonlinear and non-stationary data due to its excellent adaptivity and convergence [30]. Through the addition of adaptive noise, the modal effects are further diminished. This algorithm can decompose complex time series data into intrinsic modal functions (IMFs) and a residual (Res), so as to effectively solve problems such as boundary effects and low computational efficiency that EMD [31], EEMD [32], and CEEMD [33] are prone to.

The following are the specific steps of the CEEMDAN algorithm.

x(t) is the original passenger flow time series; $\bar{IMF_{k}}(t)$ is the *k*th IMF obtained by CEEMDAN decomposition; ${\mathrm{EMD}}_{j}(*)$ represents the *j*th IMF obtained by EMD decomposition; $\beta_{k}(k=2,3,\cdots \cdots ,K)$ is a scalar coefficient that is used to adjust the signal-to-noise ratio at each stage, determining the standard deviation of the Gaussian white noise in the process; $\omega_{i}(t)(i=1,2,\cdots \cdots ,n)$ is the Gaussian white noise that adheres to the standard normal distribution.

Step 1: The acquired x(t) is utilized for the first decomposition by adding a white noise $\omega_{i}(t)$ with a signal-to-noise ratio $\beta_{0}$ to the original time series $x_{i}(t)$, as indicated in Equation (1).
(1)$$x_{i}(t)=x(t)+\beta_{0}\omega_{i}(t)$$
where *t* stands for the various time points, *i* for the *i*th addition of white noise, and *n* for all the additions of white noise.

Step 2: Use EMD to decompose $x_{i}(t)$ *n* times, then obtain $IMF_{1}^{i}(t)$. The average value is calculated using Equation (2) to obtain the first IMF of CEEMDAN. The first residual $R_{1}(t)$ is produced using Equation (3), and $EMD_{1}(*)$ represents the first IMF obtained through EMD. Theoretically, since white noise has an average value of zero, the influence of white noise can be reduced by finding the average value.
(2)$$\bar{IMF_{1}}(t)=\frac{1}{n}\sum_{i=1}^{n}IMF_{1}^{i}(t)=\frac{1}{n}EMD_{1}\left[x_{i}(t)\right]$$
(3)$$R_{1}(t)=x(t)-\bar{IMF_{1}}(t)$$

Step 3: The first IMF derived by EMD with the inclusion of white noise $\omega_{i}(t)$ and signal-to-noise ratio $\beta_{1}$ is the adaptive noise term. The first residual $R_{1}(t)$ is then combined with the adaptive noise term to create a new time series. The second IMF of CEEMDAN is then obtained by decomposing a fresh time series using Equation (4). Equation (5) is used to generate the second residual $R_{2}(t)$.
(4)$$\bar{IMF_{2}}(t)=\frac{1}{n}\sum_{i=1}^{n}EMD_{1}\left\{R_{1}(t)+\beta_{1}EMD_{1}\left[\omega_{i}(t)\right]\right\}$$
(5)$$R_{2}(t)=R_{1}(t)-\bar{IMF_{2}}(t)$$

Step 4: Repeat Step 3, adding the new adaptive noise component to the residual term to create the new time series. After that, break it down to get the *k*th IMF of CEEMDAN. Equations (6) and (7) in particular are as follows:(6)$$\bar{IMF_{k}}(t)=\frac{1}{n}\sum_{i=1}^{n}EMD_{1}\left\{R_{k-1}(t)+\beta_{k-1}EMD_{k-1}\left[\omega_{i}(t)\right]\right\}$$
(7)$$R_{k}(t)=R_{k-1}(t)-\bar{IMF_{k}}(t)$$

Step 5: The CEEMDAN algorithm reaches a conclusion when the residual term is unable to proceed with the decomposition since it does not exceed two extreme points. The last residual R(t) at that point is a clear trend term. Equation (8) links the complete IMF to the initial time series of passenger flow.
(8)$$x(t)=\sum_{k=1}^{K}\bar{IMF_{k}}(t)+R_{k}(t)$$

### 2.2. LSTM Neural Network

Long short-term memory neural network (LSTM) is a special variant of recurrent neural networks (RNN) [34]. The gating mechanism is introduced in comparison to the original RNN, and it may recognize long-term dependencies in the input data. It can address issues like gradient explosion, gradient disappearance, and the difficulty to manage long-term dependencies brought on by intricate network layers. Although URT’s passenger flow significantly varies over the short period, it still depends on changes in both the long-term and current passenger flow levels. Therefore, accurate short-term passenger flow estimates can be made using the LSTM model. Figure 1 depicts the LSTM model structure.

The forget gate, shown as *f_t_* in the architectural diagram above, determines whether the upper layer of the LSTM’s hidden cellular state is filtered. *i_t_* stands for the input gate, *C_t−_*_1_ for the cell state at the time of the previous moment, *C_t_* for the current moment, and *O_t_* for the output gate. The current input and output are represented by *x_t_* and *h_t_*, respectively. The hyperbolic tangent function is represented by the symbol tanh, and the sigmoid function is represented by *σ*. The *w_f_*, *w_i_*, *w_o_*, and *w_c_* stand for the forget gate, input gate, output gate, and weight matrix of the cell state, respectively. The offset vectors for the forget gate, input gate, output gate, and cell state are denoted by *b_f_*, *b_i_*, *b_o_*, and *b_c_*, respectively. Below is a description of each control gate’s calculating principles.

First, the candidate state value $\tilde{C}$ of the input cell at time *t* and the value of the input gate *i_t_* are calculated:(9)$$i_{t}=\sigma \left(w_{i}\cdot \left[h_{t-1},x_{t}\right]+b_{i}\right)$$
(10)$$\tilde{C}=\tanh\left(w_{c}\cdot \left[h_{t-1},x_{t}\right]+b_{c}\right)$$

The forget gate’s activation value *f_t_* is then determined at time *t*:(11)$$f_{t}=\sigma \left(W_{f}\cdot \left[h_{t-1},x_{t}\right]+b_{f}\right)$$

It is possible to determine the cell state *C_t_* at time *t* by using the values discovered in the previous two steps:(12)$$C_{t}=f_{t}\cdot C_{t-1}+i_{t}\cdot {\tilde{C}}_{t}$$

The output gate values can be derived after getting the cell state update values:(13)$$O_{t}=\sigma \left(w_{o}\cdot \left[h_{t-1},x_{t}\right]+b_{o}\right)$$
(14)$$h_{t}=O_{t}\cdot \tanh(C_{t})$$

For the LSTM model selected in this paper, the number of training iterations *K*, the learning rate *L_r_*, and the number of neurons in the LSTM hidden layer *L*_1_, *L*_2_, are four hyperparameters that have a significant impact on the algorithm’s performance. The IPSO algorithm is used to adjust and improve the LSTM model, and these four essential hyperparameters are used as features for the particle search.

### 2.3. PSO Algorithm and Improvement

A swarm intelligence optimization technique called particle swarm optimization (PSO) mimics the social behavior of animals like fish and birds [35]. Velocity and position are the only two characteristics of the particle. Each particle’s position indicates a potential resolution to the issue, and the information that describes it is provided by its position, velocity, and fitness value. Calculating a certain fitness function yields the fitness value.

PSO begins with a set of random particles and uses continual updating and iteration to locate the best solution. Each particle will choose its own position and speed throughout each iteration based on $p_{b}$ and $g_{b}$. Equations (15) and (16) are used to update the particle’s velocity and position after determining these two best values.
(15)$$v_{i}^{t+1}=wv_{iD}^{t}+c_{1}r_{1}\left(p_{b}{}_{i}^{t}-x_{i}^{t}\right)+c_{2}r_{2}\left(g_{b}{}_{i}^{t}-x_{i}^{t}\right)$$
(16)$$x_{i}^{t+1}=x_{i}^{t}+v_{i}^{t+1}$$
where $v_{i}$ is the velocity of the particle; $x_{i}$ is the particle’s position; $c_{1}$ and $c_{2}$ are the learning factors; $r_{1}$ and $r_{2}$ are the random numbers between $\left[0,1\right]$; w is the inertia weight.

PSO has been successful in many real-world applications, however the standard PSO still struggles with local optimization and has poor convergence accuracy. This study focuses on the three improvement options listed below to address the aforementioned issues.

#### 2.3.1. Improved Adaptive Inertia Weight

The weight of inertia has a major role in determining the convergence of PSO. The local optimization capability is poor but the global capability is higher when the inertia weight is high. The inverse is also accurate. Due to the wide variety of neural network parameters, it is simple to reach a local extremum when using a typical linear decreasing technique, as illustrated in Equation (17). The adaptive change inertia weight, as described in Equation (18), is used in this research to navigate around this restriction.
(17)$$\omega =\omega_{\max}-\frac{\omega_{\max}-\omega_{\min}}{t_{\max}}\times t$$
(18)$$W=0.1-\frac{N\sum_{1}^{D}g_{b}/D}{\sum_{i=1}^{N}\left[\sum_{1}^{D}p_{b}/D\right]}$$
where $\omega_{\max}$ and $\omega_{\min}$ represent the variable’s maximum and minimum values; t and $t_{\max}$ represent the current iteration’s and maximum iteration’s iterations, respectively.

The IPSO algorithm’s early stages are characterized by a modest declining trend, a powerful global search capability, and the potential for a broadly applicable solution. The diminishing trend of W is accelerated in this algorithm’s latter stages. The convergence velocity of IPSO can be accelerated after a good solution is identified in the early stage.

#### 2.3.2. Improvement of Learning Factors

The learning factors $c_{1}$ and $c_{2}$ are used to regulate the step duration and reposition the particles to reach both the local and the global ideal positions. As the iterative process moves forward in actual applications, it is typically required to adjust the $c_{1}$ value from large to tiny in order to speed up the search speed in the initial iterations and enhance the capability of global search. To help with the local refinement search in the subsequent iteration of the iteration and enhance the local search capacity, the $c_{2}$ value is changed from small to large. Typically, the PSO algorithm sets $c_{1}=c_{2}=2$. However, this falls short of what is required for real-world applications. The linear change learning factors $C_{1}$ and $C_{2}$, as shown in Equations (19) and (20), are introduced to improve the global and local search performance of PSO.
(19)$$C_{1}\left(t\right)=2.5-2\times \frac{t}{t_{\max}}$$
(20)$$C_{2}\left(t\right)=0.5+2\times \frac{t}{t_{\max}}$$

#### 2.3.3. Improvement of Velocity and Position Update Equation

By inserting a linear model of and as indicated in Equations (21) and (22), the better particle velocity update Equation (23) is created.
(21)$$P_{b}=\frac{p_{b}+g_{b}}{2}$$
(22)$$G_{b}=\frac{p_{b}-g_{b}}{2}$$
(23)$$V_{i}^{t+1}=WV_{i}^{t}+C_{1}r_{1}\left({P_{b}}_{i}^{t}-x_{i}^{t}\right)+C_{2}r_{2}\left({G_{b}}_{i}^{t}-x_{i}^{t}\right)$$

In addition, the average dimensional information conceptual Equation (24) and adaptive determination condition Equation (25) are introduced to further enhance the local and global search capability of particles by adaptively updating the particle positions using “X=X+V” and “X=WX+(1−W)V” segments.
(24)$$\delta =\frac{1}{D}\sum_{i=1}^{D}x_{i}^{t}$$
(25)$$Q_{i}=\frac{\exp\left\{f\left[x_{i}\left(t\right)\right]\right\}}{\exp\left\{\frac{1}{N}\sum_{i=1}^{N}f\left[x_{i}\left(t\right)\right]\right\}}$$
(26)$$f\left[x_{i}\left(t\right)\right]=\frac{1}{n}\sum_{t=1}^{n}{\left[{\hat{x}}_{i}\left(t\right)-x_{i}\left(t\right)\right]}^{2}$$
(27)$$X_{i}^{t+1}=\left\{\begin{array}{l} WX_{i}^{t}+\left(1-W\right)V_{i}^{t+1},Q_{i}>\delta \\ X_{i}^{t}+V_{i}^{t+1},Q_{i}<\delta \end{array}\right.$$
where δ is the average of each particle’s dimensions information; $Q_{i}$ is the ratio between the current particle’s fitness value and the population’s average fitness value; f(∗) is the fitness value of a particle. When $Q_{i}>\delta$, it implies that IPSO is in the early stages of its search or that the current particle distribution is dispersed, as opposed to the middle or late stages of its search or the concentrated current particle distribution, which are indicated by $Q_{i}<\delta$.

In summary, the IPSO algorithm finally improves Equations (15) and (16) to Equation (28).
(28)$$\left\{\begin{array}{l} \left\{\begin{array}{l} V_{i}^{t+1}=WV_{i}^{t}+C_{1}r_{1}\left({P_{b}}_{i}^{t}-x_{i}^{t}\right)+C_{2}r_{2}\left({G_{b}}_{i}^{t}-x_{i}^{t}\right)\\ X_{i}^{t+1}=WX_{i}^{t}+\left(1-W\right)V_{i}^{t+1} \end{array}\right.,Q_{i}>\delta \\ \left\{\begin{array}{l} V_{i}^{t+1}=WV_{i}^{t}+C_{1}r_{1}\left({P_{b}}_{i}^{t}-X_{i}^{t}\right)+C_{2}r_{2}\left({G_{b}}_{i}^{t}-X_{i}^{t}\right)\\ X_{i}^{t+1}=X_{i}^{t}+V_{i}^{t+1} \end{array}\right.,Q_{i}<\delta \end{array}\right.$$

### 2.4. Evaluation Indicators

#### 2.4.1. Benchmark Function

The performance of the proposed IPSO algorithm was evaluated in this study using simulated experiments using the 10 common benchmark functions shown in Table 1 [36]. The prediction model’s convergence precision increases as the test function’s optimized value ($f_{opt}$) gets nearer to zero.

#### 2.4.2. Prediction Errors

For evaluating model performance, choosing suitable performance criteria is crucial. All models used in this research are statistically evaluated using the standard deviation (SD), root mean square error (RMSE), mean absolute error (MAE), mean absolute percentage error (MAPE), correlation coefficient (R), and coefficient of determination (R^2^). The following values would correspond to the projected value and actual value: SD = 0, RMSE = 0, MAE = 0, MAPE = 0, CC = 1, and R^2^ = 1. The following is a list of the mathematical representations:(29)$$\mathrm{SD}=\sqrt{\frac{1}{n}\sum_{t=1}^{n}{\left\{\left[\hat{y}(t)-\bar{\hat{y}}(t)]-[y(t)-\bar{y}(t)\right]\right\}}^{2}}$$
(30)$$\mathrm{RMSE}=\sqrt{\frac{1}{n}\sum_{t=1}^{n}{\left[y(t)-x(t)\right]}^{2}}$$
(31)$$\mathrm{MAE}=\frac{1}{n}\sum_{t=1}^{n}\left|y(t)-x(t)\right|$$
(32)$$\mathrm{MAPE}=\frac{1}{n}\sum_{t=1}^{n}\left|\frac{y(t)-x(t)}{x(t)}\right|$$
(33)$$R=\frac{\sum_{t=1}^{n}\left[y(t)-\bar{y}(t)][x(t)-\bar{x}(t)\right]}{\sqrt{\sum_{t=1}^{n}{\left[y(t)-\bar{y}(t)\right]}^{2}}\sqrt{\sum_{t=1}^{n}{\left[x(t)-\bar{x}(t)\right]}^{2}}}$$
(34)$$R^{2}=1-\frac{\sum_{t=1}^{n}{\left[y(t)-x(t)\right]}^{2}}{\sum_{t=1}^{n}{\left[\bar{x}(t)-x(t)\right]}^{2}}$$
where n is the total number of time series samples, y(t) and x(t) are the predicted value and actual value at time t, $\bar{y}(t)$ and $\bar{x}(t)$ are the mean value of the predicted value and actual value.

#### 2.4.3. Taylor Diagram

In addition, this paper further qualitatively evaluates the performance of the prediction models through a Taylor diagram [37]. This diagram can provide a statistical assessment of how well each model matches the other in terms of its SD, RMSE, and R, as well as a simple summary of the degree of connection between simulated and observed fields. The value of R, RMSE, and SD differences between prediction models are all represented by a single point on a two-dimensional plot in a Taylor diagram. Although this diagram’s structure is generic, it is particularly helpful when assessing complex models.

### 2.5. CEEMDAN-IPSO-LSTM Model

The complexity and non-smoothness of the original passenger flow time series of URT interfere with the neural network prediction and the problems of neural network hyperparameters determined by trial-and-error with only empirical values seriously affecting the accuracy of the prediction model. In this study, we use the CEEMDAN algorithm to break down the time series data for the passenger flow, use the LSTM hyperparameters as the object of optimization, combine them with the IPSO algorithm to determine the optimal value of the LSTM hyperparameters, and build a combined CEEMDAN-IPSO-LSTM model to accurately predict the short-term passenger flow of URT systems. Figure 2 depicts the precise prediction method, and the subsequent steps are presented in the prediction process.

Step 1: Data decomposition. CEEMDAN is used to decompose passenger flow data to obtain IMFs and Res.

Step 2: A training set and a test set are created from the passenger flow sequence that was obtained from CEEMDAN decomposition.

Step 3: Construct LSTM neural network. Initialize the batch size, hidden layer unit number, gradient limit, and other parameters of LSTM.

Step 4: Initialize the IPSO parameters at random. The size of the population, the maximum number of iterations, and the size of the particles are chosen at random.

Step 5: Create the CEEMDAN-IPSO-IPSO-LSTM prediction model and build a combination prediction model; the hyperparameters (*L*_1_, *L*_2_, *L*_r_, *K*) of LSTM are computed using IPSO. If the iteration termination conditions are met, output the optimal value of LSTM hyperparameters. If it is not satisfied, make t=t+1, and repeat steps 2-5.

Step 6: Evaluate the prediction model. CEEMDAN-IPSO-IPSO-LSTM model is evaluated by the prediction error and Taylor diagram.

## 3. Results

### 3.1. Data Set

The experimental data are the inbound and outbound passenger flow data of Yangji Station of Guangzhou Metro from 1 July 2019 to 28 July 2019 from 6:15 to 23:15. The time series was smoothed by aggregating flow data into nonoverlapping 15-min intervals [38]. This resulted in 96 samples per day. Based on the above CEEMDAN-IPSO-LSTM model, the first 75% of the data were taken as the training set and the last 25% as the test set. The sliding window length was 3; that is, the data of the first 3 weeks were used to predict the next week.

Figure 3 depicts how Yangji Station’s inbound/outbound passenger flow statistics changed throughout the experiment. Additionally, because the subway station is close to sizable residential neighborhoods, commuters frequently utilize it during the working week, and significant morning and evening peak characteristics exist, which aids in improving forecast performance. The passenger flow significantly varies during the course of a single day, as shown in Figure 3. Its pattern is quite similar during the working week, with two peaks visible each day. The first inbound/outbound peak typically occurs between 7:30 and 8:45 and 7:30 and 9:30 in the morning, and the second inbound/outbound peak usually occurs between 17:15 and 19:15 and 17:45 and 19:00 in the afternoon. The passenger volume during the morning and/or afternoon peaks is often two to three times more than during off-peak times. Weekend trends diverge from weekday trends, and there are no clear morning and afternoon peaks. Between 11:00 and 19:00, there are frequently high passenger loads. In general, Saturday has a greater passenger volume than Sunday. Due to entertainment and social events, it is also observed that there is an increase in passenger traffic late on Friday and Saturday nights.

### 3.2. CEEMDAN Decomposition

The inbound passenger flow time series was divided using CEEMDAN into a total of 12 subseries with various amplitudes and frequencies, comprising 11 IMF components and a Res component, as shown in Figure 4. It is clear that when the IMF is further decomposed, it becomes less volatile and cyclical, which is consistent with the decomposed IMF’s features. IMF1 has the highest frequency and the shortest wavelength. As the wavelength rises, the frequency of IMF2 to IMF11 drops in turn. The trend term of the inbound passenger flow sequence is represented by the residual term.

### 3.3. Benchmark Function and Comparison Algorithm

Four other evolutionary algorithms (SOA [39], WOA [40], GWO [41], and PSO) were chosen for comparison with IPSO to assess the IPSO algorithm’s performance. All comparison algorithms made use of the same set of parameters to ensure fairness. The maximum number of iterations was 1000, and the population size was set at 50. Additionally, each algorithm was individually run 50 times on each benchmark function to lessen the effect of random numbers on algorithm performance.

Table 2 compares five evolutionary algorithms across ten benchmark functions. The operation results in Table 2 show that, for the identical benchmark function, the IPSO algorithm’s minimum, maximum, mean, and SD values are, for the most part, smaller than those of other algorithms. It can be seen from the operation results in Table 2 that, under the same benchmark function, the value of minimum, maximum, mean, and SD obtained by the IPSO algorithm are smaller than other algorithms, in most cases. The IPSO algorithm performs better than other algorithms in the whole iteration process, which can enable particles to gather more stably near the global optimal value and more easily find the global optimal solution.

Figure 5 displays the ideal iterative convergence curves for each benchmark function. The convergence curve of the IPSO algorithm on most benchmark functions is below that of other algorithms. It demonstrates that IPSO not only has great convergence accuracy throughout the whole search process for each specified benchmark function, but also a faster convergence speed. The IPSO algorithm’s adaptive strategy significantly enhances the efficiency of particle optimization, avoids PSO’s inefficient iteration process, and achieves a balance between local and global search.

### 3.4. CEEMDAN-IPSO-LSTM Results

The fitness function employed in this study is the best mean square error (MSE) that the LSTM could attain throughout training. The hyperparameters derived from the optimization are *L*_1_, *L*_2_, *L_r_*, and *K*, which correspond to the minimum MSE. Figure 6a depicts the error convergent curve during the training process. It was discovered that as the iteration count increased, the error of the CEEMDAN-IPSO-LSTM model soon converged. Within four iterations, the CEEMDAN-IPSO-LSTM fitness evolution curve attained the necessary precision and then maintained the ideal fitness value, demonstrating strong learning ability. The initial and final errors of CEEMDAN-IPSO-LSTM are one order of magnitude fewer than those of CEEMDAN-PSO-LSTM, and the model accuracy significantly increases. Figure 6b displays the estimated outcomes of the LSTM hyperparameters, which are *L*_1_ = 65, *L*_2_ = 173, *L_r_* = 0.007, and *K* = 60, which were optimized by PSO and IPSO.

### 3.5. Prediction Results of Inbound and Outbound Passenger Flow

The LSTM, CEEMDAN-LSTM, and CEEMDAN-PSO-LSTM models were employed for comparison testing to confirm the accuracy of the proposed CEEMDAN-IPSO-LSTM model. Figure 7 displays the outcomes of several model predictions of data on the inbound and outgoing passenger flow. As can be observed, the trend of the actual value curves, whether during the peak time or off-peak period, is largely consistent with the forecast curves derived by various models. The CEEMDAN-IPSO-LSTM model, on the other hand, correlates to a prediction curve through thorough local observation, which has greater forecast accuracy than the other models and is more similar to the real monitoring curve, indicating the CEEMDAN-IPSO-LSTM model has strong robustness.

### 3.6. Evaluation Indicators of Prediction Models

#### 3.6.1. Quantitative Analysis Based on Prediction Errors

Table 3 shows the performance of the CEEMDAN-IPSO-LSTM model comparison to other models (LSTM, CEEMDAN-LSTM, CEEMDAN-PSO-LSTM) for both inbound and outbound passenger flow data. It can be seen that the CEEMDAN-IPSO-LSTM model respectively reduces SD, RMSE, MAE, and MAPE of inbound/outbound passenger flow data concerning the whole day of month by 12~40 persons/13~35 persons, 13~44 person/12~35 persons, 6~37 persons/12~31 persons and 5.08~46.89%/6.5~35.1%, R and R^2^ respectively increased by 0.07~2.32%/0.86~3.63% and 0.13~2.19%/0.67~1.67%. At the same time, the proposed model can achieve favorable prediction results for the different periods during weekdays and also on the weekend. This demonstrates once more the higher prediction accuracy of the CEEMDAN-IPSO-LSTM model suggested in this study.

#### 3.6.2. Qualitative Analysis Based on Taylor Diagram

Additionally, a Taylor diagram was created for each model’s prediction errors in order to qualitatively assess the characteristics of how prediction errors are distributed among different prediction models. According to Figure 8, the comprehensive ranking of prediction results is as follows: LSTM < EMD-LSTM < EEMD-LSTM < CEEMD-LSTM < CEEMDAN-LSTM < EMD-PSO-LSTM < EEMD-PSO-LSTM < CEEMD-PSO-LSTM < CEEMDAN-PSO-LSTM < EMD-IPSO-LSTM < EEMD-IPSO-LSTM < CEEMD-IPSO-LSTM < CEEMDAN-IPSO-LSTM. Among the peer models, the CEEMDAN-IPSO-LSTM model has the highest accuracy and can meet the demands for accurate short-term predictions of passenger flow.

## 4. Discussion

In this paper, we verified that the CEEMDAN-IPSO-LSTM model can accurately predict short-term passenger flow of URT. The error statistics of inbound passenger flow and outbound passenger flow demonstrate that the proposed model, combining the strong noise-resistant robustness of the CEEMDAN and the nonlinear mapping of the LSTM, outperforms other models in terms of prediction performance. Compared with the single LSTM model, the CEEMDAN-IPSO-LSTM model reduce by 40 person/35 person, 44 person/35 person, 37 person/31 person, and 46.89%/35.1% in SD, RMSE, MAE, and MAPE, and increase by 2.32%/3.63% and 2.19%/1.67% in R and R^2^, respectively. The performance improvement of CEEMDAN-IPSO-LSTM for the LSTM is significantly higher than that of the other models.

Because of the sensitivity of the short-term prediction model to the original passenger flow time series, it can consider the impact of various factors on the passenger flow series. For further study, more effective pretreatment methods of noise reduction for passenger flow data should be explored and applied to further enhance the algorithm performance. The methods that could be explored include variational mode decomposition [42], synchrosqueezing wavelet transform [43], savitzky-golay filter [44], etc.

In this paper, we only analyzed a basic prediction model of LSTM. There exist some other improvements to this model. For example, the Bi-directional LSTM [45] and gated recurrent neural network [46]. Therefore, more base models with various denoising methods should be compared and analyzed, to further strengthen the applicability of the IPSO-LSTM model in passenger flow prediction.

In addition, the CEEMDAN-IPSO-LSTM model proposed in this paper is also valuable for time series prediction of other traffic flows. At the same time, the model can be further extended from one subway station to one subway line, or even to the entire subway network, to improve the accurate prediction of short-term passenger flow in the URT system.

## 5. Conclusions

There are increasing traffic pollution issues in the process of urbanization in many countries. URT is low-carbon and widely regarded as an effective way to solve such problems. The accurate prediction of short-term passenger flow in URT systems can improve the efficiency of transport infrastructure and vehicles, and provide reference for the development of low-carbon transportation. In this study, a short-term passenger flow prediction model for URT was proposed based on CEEMDAN-IPSO-LSTM, including the framework design of CEEMDAN-IPSO-LSTM and the determination of model parameters, which successfully addresses the issues of easy local optimum fall-off, slow late convergence, and early convergence in the conventional PSO algorithm. The experimental findings showed that the CEEMDAN-IPSO-LSTM model beat other comparison models in terms of overall performance. Specifically, the CEEMDAN-IPSO-LSTM model respectively reduced SD, RMSE, MAE, and MAPE of inbound/outbound passenger flow data concerning the whole day of month by 12~40 person/13~35 person, 13~44 person/12~35 person, 6~37 person/12~31 person and 5.08~46.89%/6.5~35.1%, R and R^2^ respectively increased by 0.07~2.32%/0.86~3.63% and 0.13~2.19%/0.67~1.67%. At the same time, the proposed model achieved favorable prediction results during weekdays and at the weekend. In summary, this research validates the applicability and robustness of the CEEMDAN-IPSO-LSTM model in the area of predicting short-term passenger flow for URT systems, and extends the use of ensemble learning technology.

However, there are still a number of restrictions in this study. For instance, the current case study examined the station’s passenger flow statistics, but did not address the relationships between other lines, nor did investigate how service interruptions and spatiotemporal impacts can affect passenger flow. Additionally, multi-source data pertaining to factors such as weather, traffic, and accidents might be investigated in the future. Further research into the proposed model’s applicability to other spatial-temporal data mining applications, such trajectory prediction, would also be interesting.

## Figures and Tables

**Figure 1 ijerph-19-16433-f001:**
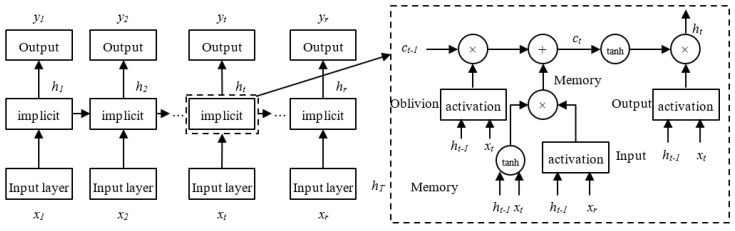
LSTM structure diagram.

**Figure 2 ijerph-19-16433-f002:**
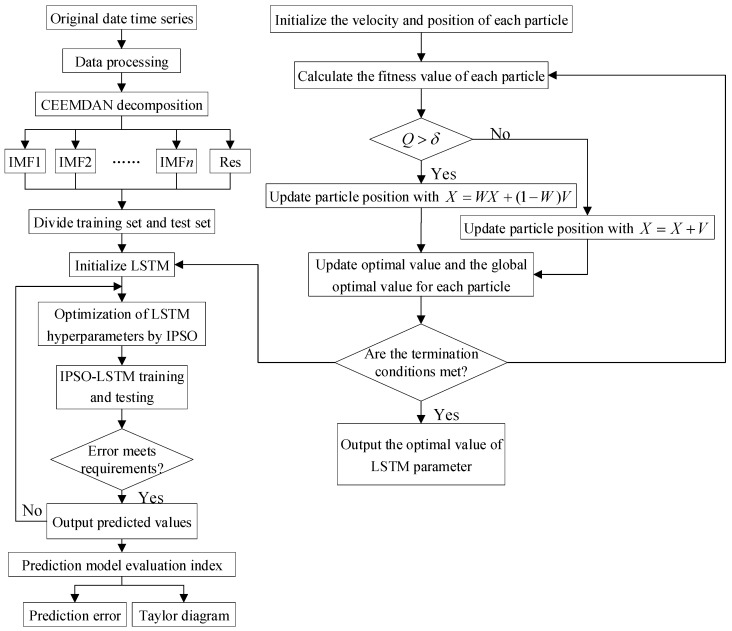
Flowchart of CEEMDAN-IPSO-LSTM prediction model.

**Figure 3 ijerph-19-16433-f003:**
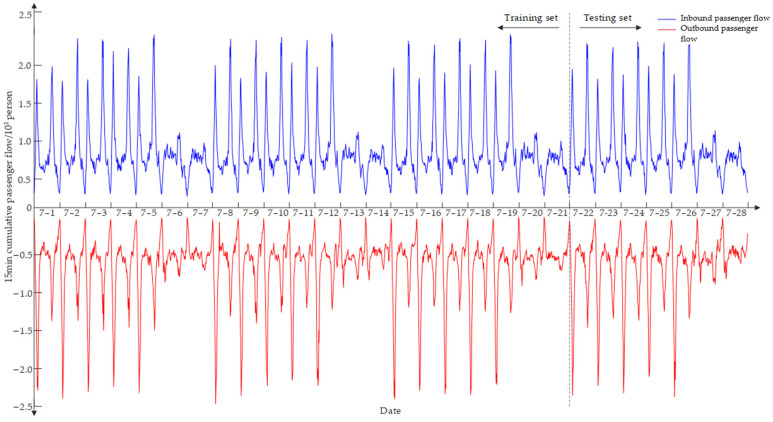
Four weeks of inbound and outbound passenger flow data.

**Figure 4 ijerph-19-16433-f004:**
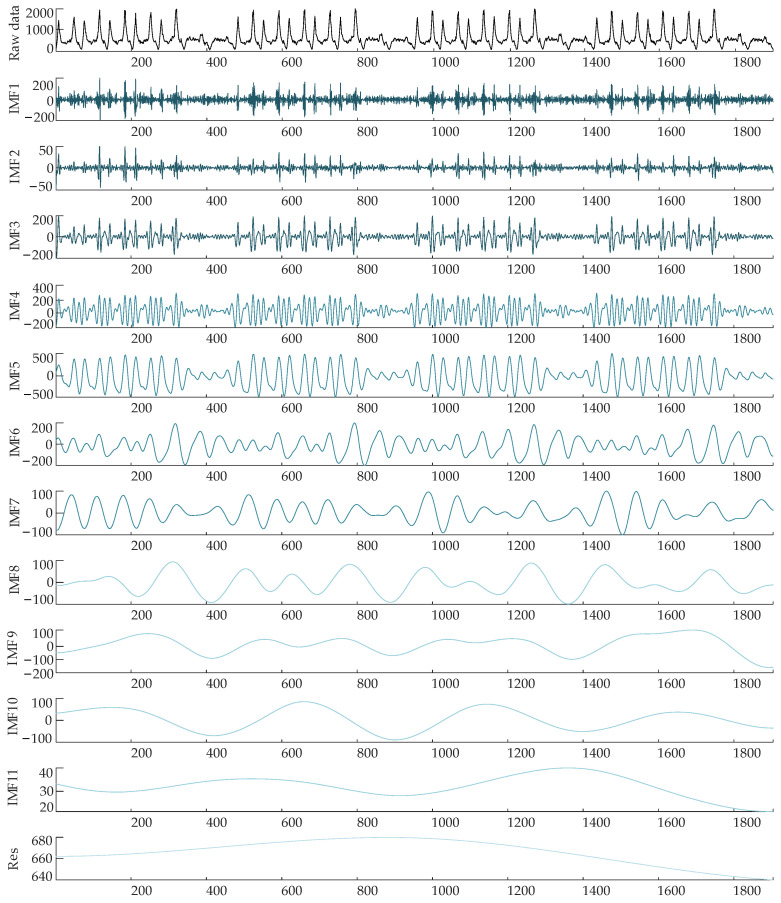
IMFs and Res obtained from the daily inbound passenger flow data after CEEMDAN decomposition.

**Figure 5 ijerph-19-16433-f005:**
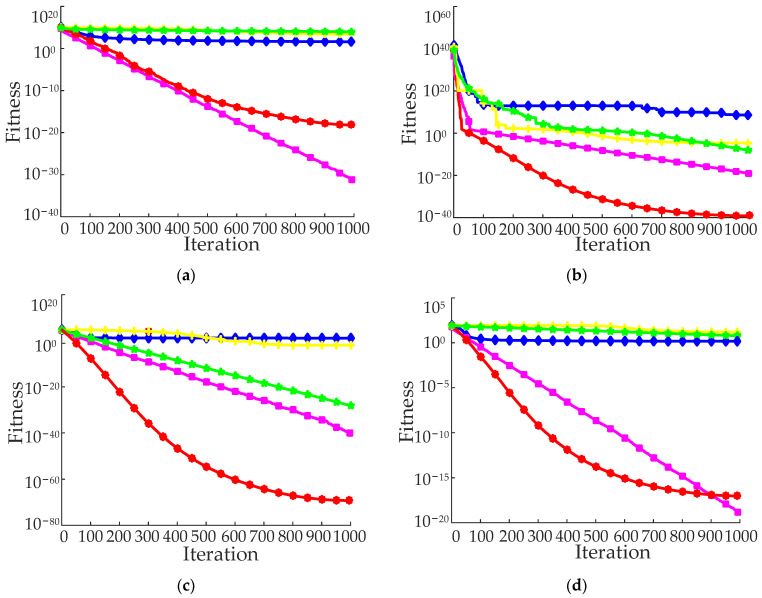
Average convergence curves of 10 benchmark functions. (**a**–**j**) represent the average convergence curve of function *f*_1_–*f*_10_.

**Figure 6 ijerph-19-16433-f006:**
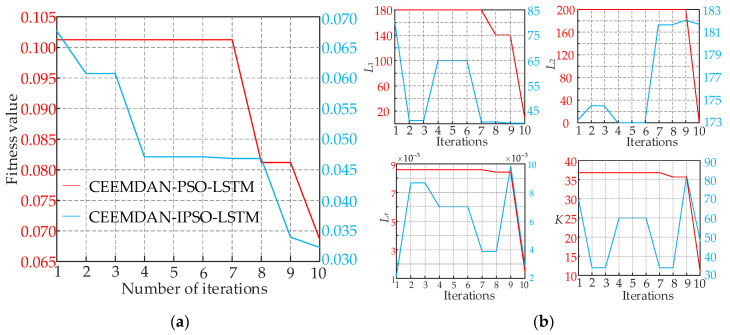
(**a**) The error convergence curves of CEEMDAN-PSO-LSTM model and CEEMDAN-IPSO-LSTM model in the training process. (**b**) The hyperparameters optimization results of CEEMDAN-PSO-LSTM model and CEEMDAN-IPSO-LSTM model.

**Figure 7 ijerph-19-16433-f007:**
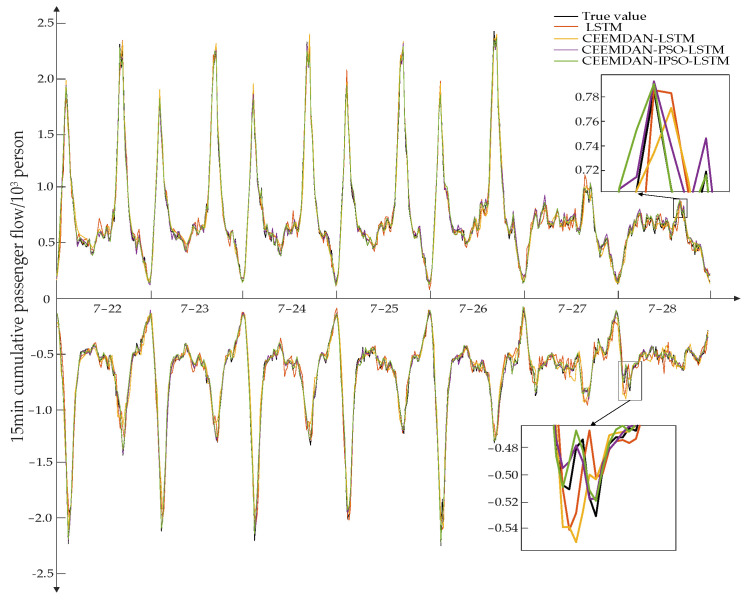
Prediction results of inbound and outbound passenger flow of different models in the last week.

**Figure 8 ijerph-19-16433-f008:**
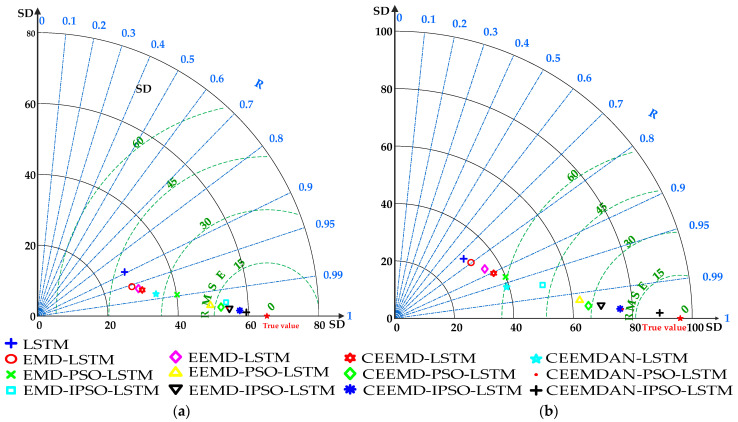
Taylor diagram of 13 prediction models. (**a**) represents the inbound passenger flow, (**b**) represents the outbound passenger flow.

**Table 1 ijerph-19-16433-t001:** Benchmark functions.

Function	Formulation	Range	$f_{opt}$
Sphere	$f_{1}\left(x\right)=\sum_{i=1}^{D}x_{i}^{2}$	$\left[-100,100\right]$	0
Sum Squars	$f_{2}\left(x\right)=\sum_{i=1}^{D}{\left(ix_{i}\right)}^{2}$	$\left[-5.21,5.21\right]$	0
Sum of Different Power	$f_{3}\left(x\right)=\sum_{i=1}^{D}{\left|x_{i}\right|}^{i+1}$	$\left[-1,1\right]$	0
Rosenbrock	$f_{4}\left(x\right)=\sum_{i=1}^{D}\left[100{\left(x_{i+1}-x_{i}^{2}\right)}^{2}+{\left(x_{i}-1\right)}^{2}\right]$	$\left[-30,30\right]$	0
Quartic	$f_{5}\left(x\right)=\sum_{i=1}^{D}{\left(ix_{i}\right)}^{2}+random\left(0,1\right)$	$\left[-1.28,1.28\right]$	0
Rastigrin	$f_{6}\left(x\right)=\sum_{i=1}^{D}\left[x_{i}^{2}-10\cos\left(2\pi x_{i}\right)+10\right]$	$\left[-5.21,5.21\right]$	0
Ackley	$f_{7}\left(x\right)=-20\exp\left(0.2\sqrt{\frac{1}{D}\sum_{i=1}^{D}{\left(x_{i}\right)}^{2}}\right)-\exp\left(\frac{1}{D}\sum_{i=1}^{D}\cos\left(2\pi x_{i}\right)\right)+20+e$	$\left[-32,32\right]$	0
Griewank	$f_{8}\left(x\right)=\frac{1}{4000}\sum_{i=1}^{D}x_{i}^{2}-\prod_{i=1}^{D}\cos\left(\frac{x_{i}}{\sqrt{i}}\right)+1$	$\left[-600,600\right]$	0
Penalized	$\begin{array}{l} f_{9}\left(x\right)=\frac{\pi }{D}\left\{10\sin^{2}\left(\pi y_{1}\right)+\sum_{i=1}^{D-1}{\left(y_{i}-1\right)}^{2}\left[1+10\sin^{2}\left(\pi y_{i+1}\right)\right]+{\left(y_{n}-1\right)}^{2}\right\}\\ +\sum_{i=1}^{D}U\left(x_{i},10,100,4\right),y_{i}=1+\frac{1}{4}\left(x_{i}+1\right),U\left(x_{i},a,k,m\right)=\left\{\begin{array}{l} k{\left(x_{i}-a\right)}^{m},x_{i}>a\\ 0,-a\leq x_{i}\leq a\\ k{\left(-x_{i}-a\right)}^{m},x_{i}<-a \end{array}\right. \end{array}$	$\left[-50,50\right]$	0
Penalized2	$\begin{array}{l} f_{10}\left(x\right)=\frac{\pi }{D}\left\{10\sin^{2}\left(\pi y_{1}\right)+\sum_{i=1}^{D-1}{\left(y_{i}-1\right)}^{2}\left[1+10\sin^{2}\left(\pi y_{i+1}\right)\right]+{\left(y_{n}-1\right)}^{2}\right\}\\ +\sum_{i=1}^{D-1}U\left(x_{i},5,100,4\right),y_{i}=1+\frac{1}{4}\left(x_{i}+1\right),U\left(x_{i},a,k,m\right)=\left\{\begin{array}{l} k{\left(x_{i}-a\right)}^{m},x_{i}>a\\ 0,-a\leq x_{i}\leq a\\ k{\left(-x_{i}-a\right)}^{m},x_{i}<-a \end{array}\right. \end{array}$	$\left[-50,50\right]$	0

**Table 2 ijerph-19-16433-t002:** Comparison results between IPSO and other evolutionary algorithms.

Function	Value	SOA	WOA	GWO	PSO	IPSO
$f_{1}$	Min	3.43 × 10^−3^	4.59 × 10^−13^	5.38 × 10^−21^	1.12 × 10	1.83 × 10^−35^
	Max	6.33 × 10	3.78 × 10^−12^	1.96 × 10^−17^	1.79 × 10	1.51 × 10^−32^
	Mean	5.21 × 10^0^	1.54 × 10^−12^	1.49 × 10^−18^	1.49 × 10	2.70 × 10^−33^
	Std	1.26 × 10	8.83 × 10^−13^	3.71 × 10^−18^	1.46 × 10^0^	3.62 × 10^−33^
	Rank	5	3	2	4	1
$f_{2}$	Min	3.43 × 10^−3^	3.14 × 10^−10^	5.38 × 10^−13^	1.42 × 10	6.06 × 10^−21^
	Max	6.33 × 10	1.30 × 10^−9^	2.88 × 10^−9^	1.78 × 10	3.11 × 10^−19^
	Mean	5.21 × 10^0^	6.15 × 10^−10^	4.56 × 10^−10^	1.59 × 10	7.02 × 10^−20^
	Std	1.26 × 10	2.19 × 10^−10^	7.18 × 10^−10^	9.45 × 10^−1^	6.26 × 10^−20^
	Rank	4	3	2	5	1
$f_{3}$	Min	1.59 × 10^3^	1.15 × 10^4^	4.11 × 10^−18^	4.23 × 10	2.09 × 10^−11^
	Max	1.40 × 10^3^	2.34 × 10^4^	1.73 × 10^−12^	8.92 × 10	7.04 × 10^−7^
	Mean	6.69 × 10^3^	1.62 × 10^4^	6.59 × 10^−14^	6.82 × 10	6.57 × 10^−8^
	Std	3.65 × 10^3^	2.85 × 10^4^	2.19 × 10^−10^	1.15 × 10	1.43 × 10^−7^
	Rank	4	5	1	3	2
$f_{4}$	Min	9.74 × 10^0^	1.25 × 10	1.41 × 10^−11^	1.31 × 10^0^	1.61 × 10^−9^
	Max	3.96 × 10	2.68 × 10	7.01 × 10^−9^	1.71 × 10^0^	6.49 × 10^−8^
	Mean	2.32 × 10	2.00 × 10	1.37 × 10^−9^	1.55 × 10^0^	1.83 × 10^−8^
	Std	9.64 × 10^0^	3.63 × 10^0^	1.68 × 10^−9^	9.59 × 10^−2^	1.53 × 10^−8^
	Rank	5	4	2	3	1
$f_{5}$	Min	1.00 × 10^−4^	9.19 × 10	1.43 × 10	1.76 × 10^2^	0.00 × 10^0^
	Max	1.26 × 10^2^	1.87 × 10^2^	1.44 × 10^2^	2.57 × 10^2^	2.07 × 10^0^
	Mean	3.01 × 10	1.46 × 10^2^	6.34 × 10	2.21 × 10^2^	6.91 × 10^−2^
	Std	2.99 × 10	2.64 × 10	3.09 × 10	1.56 × 10	3.79 × 10^−1^
	Rank	5	3	2	4	1
$f_{6}$	Min	7.18 × 10	2.47 × 10	2.85 × 10	1.66 × 10^3^	2.54 × 10
	Max	5.02 × 10^4^	4.56 × 10	2.87 × 10	5.20 × 10^3^	2.79 × 10
	Mean	7.00 × 10^3^	2.72 × 10	2.87 × 10	3.20 × 10^3^	2.66 × 10
	Std	1.21 × 10^4^	3.58 × 10^0^	3.20 × 10^−2^	9.90 × 10^2^	7.18 × 10^−1^
	Rank	5	4	3	2	1
$f_{7}$	Min	4.29 × 10^0^	3.66 × 10^−13^	1.57 × 10^−2^	1.07 × 10	2.05 × 10^−5^
	Max	1.79 × 10	2.98 × 10^−12^	1.39 × 10^−1^	1.81 × 10	1.01 × 10^0^
	Mean	7.75 × 10^0^	1.52 × 10^−12^	6.56 × 10^−2^	1.53 × 10	4.64 × 10^−1^
	Std	3.34 × 10^0^	7.21 × 10^−13^	3.03 × 10^−2^	1.99 × 10^0^	3.24 × 10^−1^
	Rank	5	4	3	2	1
$f_{8}$	Min	5.24 × 10^0^	3.34 × 10^−2^	2.46 × 10^−5^	1.62 × 10^2^	2.37 × 10^−4^
	Max	2.89 × 10^5^	8.20 × 10^−2^	7.11 × 10^−4^	4.18 × 10^2^	2.62 × 10^−3^
	Mean	4.69 × 10^4^	5.33 × 10^−2^	2.40 × 10^−4^	2.65 × 10^2^	1.31 × 10^−3^
	Std	7.42 × 10^4^	1.40 × 10^−2^	1.86 × 10^−4^	6.37 × 10	5.97 × 10^−4^
	Rank	5	4	3	2	1
$f_{9}$	Min	−1.12 × 10^3^	−1.66 × 10^3^	−1.30 × 10^3^	−1.52 × 10^3^	−1.49 × 10^3^
	Max	−7.88 × 10^2^	−1.33 × 10^3^	−9.10 × 10^2^	−9.46 × 10^2^	−1.08 × 10^3^
	Mean	−9.34 × 10^2^	−9.34 × 10^2^	−1.09 × 10^3^	−1.19 × 10^4^	−1.24 × 10^3^
	Std	7.27 × 10	7.97 × 10	1.03 × 10^2^	1.47 × 10^2^	8.49 × 10
	Rank	2	1	5	3	4
$f_{10}$	Min	3.34 × 10^−1^	1.00 × 10^−22^	5.00 × 10^−4^	2.02 × 10^−1^	0.00 × 10^0^
	Max	2.53 × 10	2.31 × 10^−19^	4.85 × 10^−1^	7.98 × 10^0^	5.23 × 10^−2^
	Mean	2.30 × 10^0^	2.76 × 10^−20^	6.85 × 10^−2^	1.37 × 10^0^	2.18 × 10^−2^
	Std	5.49 × 10^0^	5.08 × 10^−20^	1.41 × 10^−1^	1.81 × 10^0^	1.26 × 10^−2^
	Rank	5	4	2	3	1
Total Rank		45	35	25	31	14
Final Rank		5	4	2	3	1

**Table 3 ijerph-19-16433-t003:** Comparison of prediction errors.

Period		Error	Inbound				Outbound			
			L ^1^	C-L ^1^	C-P-L ^1^	C-IP-L ^1^	L ^1^	C-L ^1^	C-P-L ^1^	C-IP-L ^1^
Month	Day	SD	67	55	39	27	89	83	67	54
		RMSE	69	55	38	25	89	84	66	54
		MAE	57	41	26	20	66	60	47	35
		MAPE	82.36	73.19	40.55	35.47	83.68	80.11	55.08	48.58
		R	97.24	98.15	99.49	99.56	95.16	96.65	97.93	98.79
		R^2^	97.69	99.10	99.75	99.88	97.57	97.87	98.57	99.24
Weekday	Day	SD	62	49	33	25	86	78	45	37
		RMSE	61	47	33	24	87	78	45	37
		MAE	44	36	24	19	63	56	42	36
		MAPE	73.34	65.84	40.02	35.33	72.04	70.53	57.82	40.68
		R	98.81	99.17	99.49	99.70	96.34	97.04	98.02	99.33
		R^2^	98.34	99.42	99.74	99.86	98.21	98.57	99.52	99.67
	Peak	SD	124	112	86	78	199	151	124	100
		RMSE	123	110	87	75	203	151	126	101
		MAE	91	83	67	52	159	112	91	83
		MAPE	63.17	58.84	47.37	32.01	61.77	57.84	47.02	41.63
		R	88.33	94.54	96.24	98.7	74.20	80.66	90.12	94.73
		R^2^	93.98	94.73	97.53	98.35	87.01	93.40	95.11	97.24
	Off-Peak	SD	43	41	32	26	69	59	41	32
		RMSE	43	40	33	27	70	57	41	32
		MAE	33	29	23	20	50	43	30	25
		MAPE	72.82	62.72	50.49	47.73	70.25	68.56	56.14	52.08
		R	95.11	95.80	97.28	96.79	87.98	91.77	95.73	97.39
		R^2^	97.54	97.92	98.70	99.43	93.91	95.80	97.97	98.69
Weekend	Day	SD	116	92	60	41	120	99	73	52
		RMSE	119	93	60	42	118	100	73	55
		MAE	87	71	50	46	89	75	50	42
		MAPE	37.28	26.27	18.71	15.00	43.31	40.19	35.71	32.00
		R	79.88	83.48	88.37	92.84	73.88	80.48	89.35	91.84
		R^2^	84.06	89.64	93.18	96.88	86.29	90.64	94.18	97.59

^1^ The names of LSTM, CEEMDAN-LSTM, CEEMDAN-PSO-LSTM, and CEEMDAN-IPSO-LSTM models are abbreviated as L, C-L, C-P-L, and C-IP-L in Table 3.

## Data Availability

The datasets used and/or analyzed during the current study are available from the corresponding author upon reasonable request.

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
