# Peer review of "CEEMDAN-IPSO-LSTM: A Novel Model for Short-Term Passenger Flow Prediction in Urban Rail Transit Systems"

_ijerph, 2022, doi:10.3390/ijerph192416433_

Round 1
Reviewer 1 Report
We thank the authors of the manuscript for an interesting and relevant research. We think that under the conditions of intensively complicating transport logistics, new research is needed to take into account the nonlinear dynamics of the development of transport support for various requirements.
The use of a set of methods for assessing and forecasting traffic congestion can improve the efficiency of transport infrastructure and means of transport, thereby, as the authors point out, reduce the risks of infectious diseases due to excessive crowding of passengers, as well as reduce the negative impact on the environment through the optimization of traffic flows.
Nevertheless, in our opinion, the authors have quite strong hopes for the methods used. At the same time, it should be noted that the work has a rather narrow character. It is limited to the study of quite specific transport nodes. This in some way reduces its scientific value. However, this can be corrected in the following way. Thus, the authors decided not to discuss the results obtained (formally, there is no "Discussion" section in the work) and stop at indicating some limitations of the study (in the "Conclusions" section). Nevertheless, traffic flows are very diverse. This is obvious not only for different cities in China, but also for cities in other countries. The existence of publications of similar studies makes it possible to discuss the results obtained in the light of the achievements of earlier studies. This would allow us to understand how universal the algorithms and sequences proposed by the authors for calculating the optimal passenger flow are in terms of, firstly, organizing the schedules of means of transport, and secondly, planning their own trips directly by the users.
Many references to sources are missing from the text. I could not find references to sources 10 through 25.
There probably needs to be more emphasis on health and pollution problems as they are addressed through the methods the authors propose. These problems are at the heart of the journal.
Good luck with further research!
Author Response
Response to Reviewer 1 Comments
Point 1: There is no "Discussion" section in the work. The proposed model is limited to the study of quite specific transport nodes. This in some way reduces its scientific value.
Response 1: We gratefully appreciate for your valuable suggestion. We have added a new discussion in Section 4, and explained that the model proposed in this paper is also very useful for time series prediction of other traffic flows. At the same time, the model can be further extended from one subway station to one subway line or even the entire subway network to improve the accurate prediction of short-term passenger flow in URT system.
Point 2: Many references to sources are missing from the text. I could not find references to sources 10 through 25.
Response 2: Thank you so much for your careful check. For your convenience, we have carefully checked the citation format of all references according to the submission requirements.
Point 3: There probably needs to be more emphasis on health and pollution problems as they are addressed through the methods the authors propose. These problems are at the heart of the journal.
Response 3: We agree with you. In the abstract and introduction section, we have strengthened the description of the significance of accurate prediction results of short-term passenger flow for public safety and environmental protection.

Reviewer 2 Report
Major Revision
In order to improve the accurately anticipate the short-period term memory passenger flow of urban rail transit, this paper offers a short-term passenger flow prediction model based on complete ensemble empirical mode decomposition with adaptive noise and long-short term memory neural network. However, the manuscript could be further improved to meet the standards of the journal.
1.The introduction of this paper does not make clear the relationship between short-term passenger flow prediction and public safety as well as that between short-term passenger flow prediction and environmental protection.
2. The model does not specify the time granularity of the data and ignores the influence of time granularity on the accuracy of the model.
3. The title of this paper is the short-term passenger flow prediction of urban rail transit system, but the application case is a single rail transit station, ignoring the difference between rail station and the whole rail network passenger flow prediction, whether the model conclusion is applicable to the rail network needs to be further explained.
Author Response
Response to Reviewer 2 Comments
Point 1: The introduction of this paper does not make clear the relationship between short-term passenger flow prediction and public safety as well as that between short-term passenger flow prediction and environmental protection.
Response 1: Your suggestion is valuable. In abstract and introduction section, we have strengthened the description of the significance of accurate prediction results of short-term passenger flow for public safety and environmental protection.
Point 2: The model does not specify the time granularity of the data and ignores the influence of time granularity on the accuracy of the model.
Response 2: We are very sorry for your misunderstanding caused by the mistake of forgetting to specify the time granularity of the experimental data. We have referred to the research results in Reference [38] and counted the passenger flow with the time granularity of 15min. The order of the following references is adjusted.
Point 3: The title of this paper is the short-term passenger flow prediction of urban rail transit system, but the application case is a single rail transit station, ignoring the difference between rail station and the whole rail network passenger flow prediction, whether the model conclusion is applicable to the rail network needs to be further explained.
Response 3: We are sorry for our incomplete expression, which has caused your misunderstanding. In order to improve the structure of the article, we added a new discussion in Section 4, and explained that the model proposed in this paper is also very useful for time series prediction of other traffic flows.At the same time, the model can be further extended from one subway station to one subway line or even the entire subway network to improve the accurate prediction of short-term passenger flow in URT system.

Reviewer 3 Report
Authors put forward a prediction method based on CEEMDAN-IPSO-LSTM, which solves the problem of short-term passenger flow prediction of urban rail transit. The research is meaningful, but there are the following parts that can be revised:
1. line11-line15, "Urban rail transit is... transport resources", the background in the abstract is too long, it should be appropriately simplified and cleared about the research object.
2. line24-26, "Third, the empirical findings demonstrated..." , it should quantitatively describe the advantages of the method.
3. line83, the purpose and significance of the study should be mentioned at the end of the introduction.
4. All variables in the formula are not completely explained, please check carefully.
5. In section 3.2, authors use CEEMDAN algorithm to decompose the passenger flow time series. Was the validity of this decomposition (Signal to noise, Retention energy ratio, Standard deviation, etc.) tested?
6. Authors should check all details in the manuscript, such as wrong format (Line330, line362, line436), awkward words, etc.
Author Response
Response to Reviewer 3 Comments
Point 1: line11-line15, "Urban rail transit is... transport resources", the background in the abstract is too long, it should be appropriately simplified and cleared about the research object.
Response 1: We gratefully appreciate for your valuable suggestion. We have made it clear in the part of abstract that the research object in this paper is the short-term passenger flow of urban rail transit, and the description of the background is simplified. The specific changes are in line11-13 of the new manuscript.
Point 2: line24-26, "Third, the empirical findings demonstrated..." , it should quantitatively describe the advantages of the method.
Response 2: Yes, the quantitative description of prediction results of the model is helpful to explain the prediction effect. We have modified as you suggested. The specific changes are in line23-26 of the new manuscript.
Point 3: line83, the purpose and significance of the study should be mentioned at the end of the introduction.
Response 3: We have adjusted the purpose and significance of this study to the last part of the introduction according to your suggestions. And it looks much better than before! The specific changes are in line82-86 of the new manuscript.
Point 4: All variables in the formula are not completely explained, please check carefully.
Response 4: Thank you so much for your careful check. The interpretation of variables in all formulas in this paper is involved in the context.
Point 5: In section 3.2, authors use CEEMDAN algorithm to decompose the passenger flow time series. Was the validity of this decomposition (Signal to noise, Retention energy ratio, Standard deviation, etc.) tested?
Response 5: The focus of this paper is to use IPSO algorithm to solve the optimal value of syperparameters of LSTM neural network. CEEMDAN algorithm is only used as a method to remove data noise before prediction of IPSO-LSTM model. Moreover, detailed principles and tests are presented in reference [30]. Therefore, from our point of view, the effectiveness of CEEMDAN decomposition is valuable but not necessary for the main purpose of this paper.
Point 6: Authors should check all details in the manuscript, such as wrong format (Line330, line362, line436), awkward words, etc.;
Response 6: Considering the language and grammar problems, we have carefully checked all the details in the manuscript, including wrong format,awkward words, etc.

Round 2
Reviewer 1 Report
Dear colleagues!
Thank you for your work. Your manuscript is now better.
But again I did not find references to sources (10-25) in the text of the manuscript. They need to be inserted in the text.